



Calcification in a marginal sea – influence of seawater $[Ca^{2+}]$ and carbonate chemistry on
bivalve shell formation
Jörn Thomsen[1], Kirti Ramesh[,1,2], Trystan Sanders[1], Markus Bleich[2], Frank Melzner[1]
[1]Marine Ecology, GEOMAR Helmholtz Centre for Ocean Research, Kiel, Germany
[2]Institute of Physiology, Christian-Albrechts-University Kiel, 24098 Kiel, Germany
Running headline: Abiotic effects on mussel calcification
**Abstract**
In estuarine coastal systems such as the Baltic Sea, mussels suffer from low salinity which
limits their distribution. Anthropogenic climate change is expected to cause further
desalination which will lead to local extinctions of mussels in the low saline areas. It is
commonly accepted that mussel distribution is limited by osmotic stress. However, along the
salinity gradient environmental conditions for biomineralization are successively becoming
more adverse as a result of reduced $[Ca^{2+}]$ and dissolved inorganic carbon ($C_T$) availability.
In larvae, calcification is an essential process starting during early development with
formation of the prodissoconch I (PD I) shell which is completed under optimal conditions
within 2 days.
Experimental manipulations of seawater $[Ca^{2+}]$ start to impair PD I formation in *Mytilus* larvae
at concentrations below 3 mM, which corresponds to conditions present in the Baltic at
salinities below 8 g $kg^{-1}$. In addition, lowering dissolved inorganic carbon to critical
concentrations (<1 mM) similarly affected PD I size which was well correlated with calculated
$\Omega_{Aragonite}$ and $[Ca^{2+}][HCO_3^-]/[H^+]$ in all treatments. Comparing results for larvae from the
western Baltic with a population from the central Baltic revealed significantly higher tolerance
of PD I formation to lowered $[Ca^{2+}]$ and $[Ca^{2+}][HCO_3^-]/[H^+]$ in the low saline adapted
population. This may result from genetic adaptation to the more adverse environmental
conditions prevailing in the low saline areas of the Baltic.
The combined effects of lowered $[Ca^{2+}]$ and adverse carbonate chemistry represent major
limiting factors for bivalve calcification and can thereby contribute to distribution limits of
mussels in the Baltic Sea.
**Key-words**
Baltic Sea, bivalves, calcium, calcification, carbonate chemistry, climate change
**1. Introduction**
Salinity is one of the most important environmental parameters limiting the distribution of
aquatic species. Many marine organisms exhibit little tolerance to reduced salinity and are
thus not able to thrive in brackish water environments influenced by riverine inputs (Whitfield
et al. 2012). On the other hand, some animals, such as bivalves and crustaceans tolerate the
dilution of the ambient seawater and are able to inhabit estuarine, brackish water habitats
(Westerbom et al. 2002). However, within these habitats, organisms need to tolerate a
number of environmental stressors which are changing concomitantly.
Generally, lowered ambient ion concentrations affect an organism's ability to maintain
cellular homeostasis. In response, some organisms such as crustaceans actively regulate
the ionic composition of their extracellular fluids. However, mytilid mussels do not control
haemolymph osmolarity and ionic composition mostly corresponds to that of ambient
seawater (Thomsen et al. 2010). Thus tissues are subjected to a diluted medium in brackish
water but the inorganic composition of the intracellular space needs to be regulated in order
to maintain enzymatic functions. At moderately lowered salinity, intracellular $[K^+]$ and $[Na^+]$
are kept relatively stable at about 200 and 100 mM, respectively, but $[K^+]$ drops rapidly under
strong hypoosmotic stress to avoid cell swelling (Willmer 1978, Wright et al. 1989; Silva and
Wright 1994). In order to stay iso-osmotic with their environment following long-term
acclimation to lowered salinity, intracellular $[K^+]$ and $[Na^+]$ are maintained at lower
concentrations (Willmer 1978, Natochin et al. 1979). In addition, bivalves reduce the



concentration of intracellular compatible organic osmolytes (Hochachka and Somero 2002)
such as certain amino acids, taurine and betaine during the acclimation phase (Silva and
Wright 1994, Kube et al. 2006). However, at a certain critical salinity threshold ($S_{crit}$), the
intracellular organic osmolyte pools are depleted which has been suggested to eventually
limit species fitness (Kube et al. 2006; Podbielski et al. 2016).
At the same time, bivalves produce an external shell composed of $CaCO_3$ and an organic
matrix (Falini et al. 1996). The shell enables adult bivalves to live in intertidal habitats and is
an effective protection against predation but shell formation has been shown to be sensitive
to lowered salinity (Malone and Dodd 1967). Under favourable environmental conditions,
calcification begins already in early development and the first larval shell (prodissoconch I,
PD I) is completed within the first 48 hours after fertilization. PD I formation is an important
prerequisite for the successful development of bivalve larvae as larvae seem to commence
feeding only after completion of the shell which provides structural support (e.g. muscle
attachment site) for the functional velum (Lucas and Rangel 1983; Cragg 1985). However,
PD I formation is highly sensitive to chemical and environmental stressors (Williams and Hall
1999) and initiation of feeding is delayed under adverse carbonate chemistry (Waldbusser et
al. 2015).
Recently, a number of studies investigated how changes of seawater carbonate chemistry
affect marine calcifiers. Those studies were mostly motivated by the ongoing input of
anthropogenic $CO_2$ into the oceans which results in a drop of pH and lowered [$CO_3^{2-}$], a
process called ocean acidification. Bivalve shell formation is highly sensitive to modifications
of carbonate chemistry and therefore negatively affected by ocean acidification (Gazeau et al.
2013; Waldbusser et al. 2014; Thomsen et al. 2015). The exact reason for the sensitivity of
calcification to adverse carbonate chemistry is still under debate (Cyronak et al. 2015).
Lowered saturation of seawater with respect to calcium carbonate ($\Omega$, [$Ca^{2+}$][$CO_3^{2-}$]/ K*sp)
(with K*sp=stoichiometric solubility product (Mucci 1983)) could affect the kinetic of shell
formation (according to r = $k(\Omega-1)^n$ with r=mineral precipitation rate, k=rate constant and
n=reaction order, Waldbusser et al. 2014) and undersaturation leads to dissolution of existing
calcium carbonate structures (Thomsen et al. 2010; Melzner et al. 2011, Haynert et al. 2014).
Alternatively, the substrate inhibitor ratio (SIR) defined as the availability of the substrate for
calcification in the form of dissolved inorganic carbon ($C_T$) or $HCO_3^-$ and the inhibitory effect
of lowered seawater pH (increased [$H^+$]) could restrict calcification rate (Bach 2015; Thomsen
et al. 2015; Fassbender, et al. 2016).
Independent of the exact mode of action, larval bivalve calcification is driven by uptake of
seawater $Ca^{2+}$ and inorganic carbon ($C_T$) whereas metabolic carbon is only of minor
importance and contributes by less than 10 % in larvae and adults (McConnaughey and
Gillikin 2008, Waldbusser et al. 2015). Oceanic [$Ca^{2+}$] is about 10 mM, but necessarily
linearly related with seawater salinity and thus reduced in estuaries. Freshwater [$Ca^{2+}$] are in
general much lower (<1-2 mM [$Ca^{2+}$], Ohlson and Anderson 1990; Juhna and Klavins 2000).
Oceanic $C_T$ is about 2 mM whereby $HCO_3^-$ and $CO_3^{2-}$ contribute about 90 and 8 % to the $C_T$
pool, respectively. $C_T$ of seawater equilibrated with the atmosphere is directly proportional to
salinity as it is depending on seawater total alkalinity ($A_T$). Therefore, calcifiers are facing
abiotic conditions in brackish water habitats which most likely affect their ability to form a
shell.
The Baltic Sea is an example of a brackish water habitat which is substantially influenced by
precipitation and riverine input (Gustafsson et al. 2014) which results in a salinity gradient
from 25 g kg$^{-1}$ in the Kattegat transition zone to basically freshwater in the Gulfs of Riga,
Finland and Bothnia. As a consequence, [$Ca^{2+}$], $A_T$ and $C_T$ decline linearly along the salinity
gradient (Kremling and Wilhelm 1997; Beldowski et al. 2010). However, varying composition
of riverine freshwater results in differing $A_T$ -salinity correlations and in the Gulf of Riga, $A_T$
and thus $C_T$ even increases with lowered salinity (Beldowski et al. 2010).
The Baltic Sea is among the coastal ecosystems which are most heavily influenced by
anthropogenic activity. Eutrophication enhanced hypoxia or even anoxia events in the
benthic ecosystem. As respiratory oxygen consumption is coupled to $CO_2$ production,
hypoxia is always accompanied by a pronounced increase of $pCO_2$ and thus affects the



carbonate system simultaneously (Melzner et al. 2013). Furthermore, climate change is
expected to increase precipitation in the Baltic catchment area which may cause increased
riverine runoff leading to reduced salinity (0 - 45 % reduction) in particular in the north-
eastern and central Baltic Sea (Meier et al. 2006; Gräwe et al. 2013). This shift in salinity will
most likely induce a substantial retreat of the marine fauna and flora and expansion of limnic
species into the formerly brackish water habitats (Johannesson et al. 2011).
Mytilid mussels (*Mytilus* spp.) are among the most abundant organisms of the Baltic Sea
($10^{13}$ individuals) contributing up to 90% to local hard bottom biomass, and thus are
important habitat builders (Enderlein and Wahl 2004, Johannesson et al. 2011). Their
distribution along the Finish, Swedish and Estonian coast is limited by salinities of about 4.5
g kg$^{-1}$ when abundance, biomass and growth drastically decline (Westerbom et al. 2002;
Martin et al. 2013; Riisgard et al. 2014). As growth combines both somatic growth and shell
formation, it is unclear which physiological mechanism exactly limits performance and
therefore the distribution of mussels (Riisgard et al. 2014).
Currently, distribution limits of marine bivalves in estuaries are commonly related to the
inability of intracellular osmoregulatory adjustment at lowered salinity (Maar et al. 2015).
However, as [Ca$^{2+}$] and $C_T$ availability decline along the Baltic Sea salinity gradient it is likely
that the calcification process is negatively affected as well. This process has not been
previously considered as a factor contributing to distribution limits of mussels. In this study,
we investigated the effects of seawater [Ca$^{2+}$] independently of salinity in combination with
lowered $C_T$ availability on the calcification performance of larval *Mytilus* spp. and correlated
the experimental data with environmental conditions present in the Baltic Sea.
**2. Material and Methods**
2.1 Animal collection and spawning
Adult mussels were collected from subtidal depths at the pier of GEOMAR in Kiel Fjord (shell
length: 4-6 cm, 54°19.8'N; 010°09.0'E) and at the wooden groynes close to Koserow on the
island of Usedom (shell length: 2-3 cm; 54°03.4'N; 014°00.4'E) between May and June 2016
(Fig. 1). Median salinity for Kiel Fjord and Usedom, located ~350 km east of Kiel, are ~ 17
and 7 g kg$^{-1}$, respectively (Table 1).
Mussels in the Baltic Sea represent hybrids of *Mytilus edulis* x *trossulus* with increasing
*trossulus* allele frequency towards the less saline, eastern Baltic (Stuckas et al. 2009). Thus
mussels collected in Kiel represent the Baltic *M. edulis*-like and animals from Usedom belong
to the *M. trossulus*-like genotype (Stuckas et al. 2017).
Specimens were either used for spawning immediately after collection or kept in cold storage
(9°C) in order to delay gonad maturation for up to 3 months. Stored mussels (ca. 500 g
mussel wet biomass per 20 L tank, 12 tanks) were fed 6 times a week with 500 mL of
*Rhodomonas* solution (ca. 2 x 10$^6$ cells mL$^{-1}$) supplemented with a commercial bivalve diet
(Acuinuga, Spain) and water was exchanged twice a week (Thomsen et al. 2010).
*Rhodomonas* spp. were cultured in PES medium as described previously with the exception
of using 40 L cylinders (Thomsen et al. 2010).
All experiments were performed at 17°C. Spawning was induced by exposing the animals to
rapidly elevated water temperature between 18-25°C using heaters. Spawning specimens
were separated from the remaining animals and eggs and sperms were collected individually
in beakers filled with 0.2 µm filtered seawater (FSW). Subsequently, eggs were pooled and
fertilized with a pooled sperm solution. For the Kiel population, 5 individual experimental runs
were performed with varying number of dams and sires used for crossings in each run. In
total 16 dams and 18 sires were used. For the Usedom population one run with 4 replicates
was performed for which gonads from 5 dams and 4 sires were pooled. Fertilization success
was determined by verifying the presence of a polar body and first and second cell division of
zygotes and was above 90% in all runs. Embryos (4-8 cell stage) and non-calcified
trochophora (in one experimental run of the Kiel population) from all parents were transferred
in equal numbers into the experimental units (volume: 25 or 50 mL in round plastic beakers)
at a density of 10 embryos/larvae mL$^{-1}$.





Three days post fertilization animals were removed from the experimental units by filtering
the full water volume through a filter with a mesh size of 20 µm or by collecting larvae
individually using a pipette in treatments with low survival. Subsequently, larvae were fixed
using 40 % paraformaldehyde (PFA, pH 8.0) resulting in a final PFA concentration of 4%.
Pictures of larvae were taken using a stereomicroscope (Leica M165 FC) equipped with a
Leica DFC 310 FX camera and LAS V4.2 software. Calcification was assessed by measuring
the larval shell length. PD I shell length was assessed using Image J 1.50i by measuring the
maximal shell length in parallel to the hinge or the maximal shell diameter for larvae that had
not developed a complete PD I shell.
2.2 Experimental manipulation of seawater [$Ca^{2+}$] and carbonate chemistry
Artificial seawater (ASW) was prepared according to Kester (1967) for salinities of 14 and 7 g
$kg^{-1}$ for experiments with *M. edulis*-like and *trossulus*-like, respectively, by adding NaCl,
NaSO$_4$, KCl, NaHCO$_3$, KBr, H$_3$BO$_3$, MgCl$_2$, CaCl$_2$, and SrCl$_2$ to deionised water. $Ca^{2+}$ free
artificial seawater (CFSW) was prepared by omitting CaCl$_2$ and adjusting osmolarity similar to
ASW by increasing NaCl concentrations. pH$_{NBS}$ was adjusted to 8.0 using NaOH. All
experimental treatments comprised 5 % of 0.2 µm filtered seawater (FSW) from Kiel Fjord
which was adjusted to salinity 7 g $kg^{-1}$ for the Usedom population experiment to ensure that
trace elements were present. The comparison of shell sizes of larvae kept in control ASW +
5% FSW or 100 % FSW yielded no significant differences (p>0.05). Varying seawater [$Ca^{2+}$]
treatments were prepared by mixing ASW and CFSW (lowered [$Ca^{2+}$]) or by addition of CaCl$_2$
from a 500 mM stock solution to ASW (elevated [$Ca^{2+}$]). Following mixing, water samples
were taken and seawater [$Ca^{2+}$] was measured using a flame photometer (EFOX 5053,
Eppendorf, Germany) calibrated with urine standards (Biorapid GmbH, Germany).
Seawater carbonate chemistry was manipulated by increasing alkalinity by addition of
[NaHCO$_3$] to ASW or by lowering alkalinity by adding 1M HCl to the experimental units.
Excess $CO_2$ was removed by aeration of the experimental units for 30 min and embryos were
only added after pH had increased again to stable values (~7.8). Seawater pH was
determined on the NBS scale using a WTW 3310 pH meter equipped with a Sentix 81
electrode. Seawater $C_T$ was determined using an AIRICA $CO_2$ analyzer and verified by
measuring certified reference material (Dickson et al. 2003). Seawater carbonate system
parameters (HCO$_3^-$, CO$_3^{2-}$, $\Omega_{aragonite}$) were calculated using the CO2SYS program with
KHSO4, K1 and K2 dissociation constants after Dickson et al. (1990) and Roy et al. (1993),
respectively. pH$_{NBS}$ was converted to total scale pH. $\Omega_{aragonite}$ and [$Ca^{2+}$][HCO$_3^-$]/[H$^+$] were
linearly adjusted according to measured seawater [$Ca^{2+}$] (Table 2).
2.3 Microelectrode measurements of [$Ca^{2+}$] in the calcifying space of D-stage veliger
Using ion-selective electrodes, $Ca^{2+}$ gradients were measured in seawater and in the
calcification space (CS) below the surface of the shell in veliger larvae three days after
fertilization. The experimental set up and hardware was identical to that of Stumpp et al.
(2012), except for the addition of a metal plate connected to a water cooling system for
temperature control.
Borosilicate glass capillary tubes (inner diameter 1.2 mm, outer diameter, 1.5 mm) with
filament were pulled on a DMZ-Universal puller (Zeitz Instruments, Germany) to
micropipettes with tip diameters of 1-3 µm. Micropipettes were silanized with dimethyl
chlorosilane (Sigma-Aldrich, USA) in an oven at 200°C for 1h. Calcium sensitive liquid ion
exchangers (LIX) and LIX-PVC membranes were prepared according to de Beer et al. (2000)
with $Ca^{2+}$ ionophore II (Sigma Aldrich). The microelectrodes were back filled with a KCl based
electrolyte (200 mM KCl, 2 mM CaCl$_2$.2H$_2$O) and thereafter front loaded with LIX and finally
LIX-PVC at a length of 150 µm and 50 µm, respectively. To measure calcium in the CS,
larvae were placed into the temperature controlled perfusion chamber mounted on an
inverted microscope (Axiovert 135, Zeiss, Germany) at a density of 100 mL$^{-1}$ and were held
in position using a holding pipette. The ion-selective probe was mounted on a remote-
controlled micro-manipulator and was introduced beneath the shell from the side of the



growing edge, where stable measurements were obtained within 5-10 seconds.
Microelectrode calibration was verified by measuring [$Ca^{2+}$] of seawater standards as
described above and analogue outputs were channelled through an amplifier (WPI
Instruments, USA) to a chart recorder (Gould Instruments, USA).
2.4 Seawater [$Ca^{2+}$] and carbonate chemistry of the Baltic Sea
Seawater [$Ca^{2+}$] (mM $kg^{-1}$) was calculated for salinities between 3 and 20 g $kg^{-1}$ using the
correlation for chlorinities <4.5 and >4.5 g $kg^{-1}$ provided by Kremling and Wilhelm (1997) and
a salinity-chlorinity conversion after Millero (1984). [$Ca^{2+}$] was calculated for salinity values
measured in Kiel Fjord (N=4250, weekly measurements 2005-2009, 0-18 m, 54°19.8' N,
10°9.0' E, Clemmesen et al., unpublished, Casties et al. 2015) and at the Oder Bank
(N=260,000, hourly measurements, 2000-2015, 3+12 m water depths, 54°4.6' N, 14°9.6' E,
~8 km off the *M. trossulus*-like collection site at Usedom (BSH 2000-2015, Table. 1). As
distribution of mytilid bivalves is limited by salinities below 4.5 g $kg^{-1}$ the calculation covers
the full [$Ca^{2+}$] range relevant for mussels in this estuary (Westerbom et al. 2002). Carbonate
chemistry calculations are based on the salinity-alkalinity correlation published by Beldowski
et al. (2010) for salinities between 3 and 20 g $kg^{-1}$ and a seawater surface $pCO_2$ of 400 µatm
assuming equilibrium with current atmospheric $CO_2$ concentrations of ~400 ppm.
Calculations were performed for seawater temperatures of 15°C which corresponds to
average conditions experienced by larvae during the natural reproductive period from April to
June. The Baltic Sea has four sub areas which are differentially impacted by the inflow of
riverine freshwater and their respective chemical properties: the Central Baltic Sea with the
Kattegat transition area, the Gulf of Riga, the Gulf of Finland and the Bothnian Sea with Gulf
of Bothnia. Depending on the chemical properties of the riverine input, seawater carbonate
chemistry can differ substantially for similar salinity values between the four regions. The
same calculations were performed for predicting future conditions using atmospheric $CO_2$
concentration of 800 ppm.
2.5 Statistical analysis
All statistical analyses (t-test, Kruskal-Wallis test followed by Dunn's test, regression analysis,
linear and nonlinear model parameter fitting) were performed using R and the mosaic
package. Population comparisons were performed by fitting linear models for log transformed
data. Each experimental unit was considered as a replicate. Values in text and figures are
replicate means ± standard error.
**3. Results**
3.1 PD I shell formation and CS [$Ca^{2+}$] under varying seawater [$Ca^{2+}$]
Larval development until PD I formation was investigated for *M. edulis*-like collected in Kiel
Fjord. The lowest seawater [$Ca^{2+}$] tested in the experiment was 0.51 mM which did not allow
successful development of larvae to the trochophore stage in the Kiel population and was
thus not considered in subsequent experiments. At all other [$Ca^{2+}$] treatments, early
development was not adversely affected and larvae started to calcify prodissoconch I.
However, at [$Ca^{2+}$] of <2 mM larvae were not able to produce a complete PD I shell. Even
after 7 days, shell size did not increase above a mean diameter of 63.7 ± 6.0 µm although
larvae stayed viable and continued to actively swim. In all other treatments, shells were fully
developed within 72 h post fertilization, but shell length declined linearly at [$Ca^{2+}$] below 3
mM ranging between 104.5 ± 2.1 µm at 2.8 mM and 82.1 ± 1.5 µm at 1.6 mM, with significant
reductions below 2.5 mM [$Ca^{2+}$] (H: 50.3, p<0.001, Dunn's test). Specimens kept at control
[$Ca^{2+}$] of 4-5 mM had mean lengths of 108.2 ± 2.5 µm. Modifications of seawater [$Ca^{2+}$] in the
range 4-10 mM had only minor impacts on lengths and elevated [$Ca^{2+}$] did not cause a
further increase of shell lengths above control size (Fig. 2a, Table 3a).
Microelectrode measurements of [$Ca^{2+}$] in the CS of *M. edulis*-like revealed that CS [$Ca^{2+}$]
drops with seawater [$Ca^{2+}$], (H: 21.2, p<0.01, Fig. 3a). However, larvae kept at 3.5 mM [$Ca^{2+}$]
(above the critical [$Ca^{2+}$] threshold) are characterized by CS [$Ca^{2+}$] of 0.1 ± 0.01 mM above
seawater concentrations (paired t-test: t= 16.9, p<0.01, Fig. 3b). In larvae raised at 2.6 and





2.3 mM [Ca$^{2+}$], the difference between seawater and CS [Ca$^{2+}$] declined to 0.06 ± 0.03 and
0.03 ± 0.02 mM which was not significantly enriched compared to the ambient seawater. In
contrast, the gradient between CS and seawater increased to 0.28 ± 0.02 mM in larvae
grown at 1.5 mM.
Results for shell formation rates of *M. edulis*-like larvae were compared with the *M. trossulus*-
like population from Usedom. Larvae were exposed to [Ca$^{2+}$] between 0.4-5.8 mM (Fig. 1b,c).
Overall, the response curve for *M. trossulus*-like was similar to *M. edulis*-like (Table 3b).
Maximal shell sizes observed at 3.7 mM were 120 ± 1.5 µm and shell lengths started to
decline at lower [Ca$^{2+}$]. Nevertheless, at comparable [Ca$^{2+}$] shell sizes were larger compared
to *M. edulis*-like and larvae were able to calcify a full PD I even at 1.1 mM [Ca$^{2+}$] with an
average size of 81.9 ± 3.2 µm. In contrast, PD I formation was not completed at 0.4 mM, yet
larvae started to calcify. A linear model of the calcification response revealed a significant
effect of [Ca$^{2+}$] and population on shell size but no interaction (Table 4a, Fig. 2c).
3.2 Combined effects of seawater [Ca$^{2+}$] and carbonate chemistry on larval calcification
*M. edulis*-like larvae were exposed to a range of seawater [Ca$^{2+}$] between 1 and 10 mM and
$C_T$ concentrations between 880-3520 µM. PD I size was not modulated by increased
seawater $C_T$ of 2900-3520 µM compared to control conditions ($C_T$: 1773 µM) and shell length
was only negatively affected by seawater [Ca$^{2+}$] below 3 mM (Fig. 4a). In contrast, lowered
seawater $C_T$ (975 µM) significantly affected shell formation and PD I length declined to 72.5 ±
2.7 µm at control [Ca$^{2+}$]. Within these treatments shell length was marginally positively
correlated with seawater [Ca$^{2+}$] but shell length remained reduced in all [Ca$^{2+}$] treatments
(linear regression: 63 (± 2.2) µm + 2.9 (± 0.7) x [Ca$^{2+}$], F:18.6, p<0.01, R$^2$= 0.47, Fig. 4a).
Whereas, the correlation of shell length against [Ca$^{2+}$] under reduced $C_T$ differed significantly
from the three higher $C_T$ treatments. Plotting PD I sizes against seawater $\Omega_{Aragonite}$ and
[Ca$^{2+}$][HCO$_3^-$]/[H$^+$] revealed a similar correlation of calcification in all treatments (Fig. 4b, c).
Calcification of larvae started to decline at $\Omega_{Aragonite}$ below 1 with significant reductions in the
treatments with $\Omega_{Aragonite}$ below 0.5 (H: 44.5, p<0.001, Dunn's test). Similarly, PD I size
declined at [Ca$^{2+}$][HCO$_3^-$]/[H$^+$] values below 0.7 and shells were significantly smaller  below
0.3 (H:42.5, p<0.01, Dunn' test). In addition, the shell formation responses of *M. edulis*-like
and *M. trossulus*-like to combined manipulations of [Ca$^{2+}$] and carbonate chemistry were
more similar compared to the effects of lowered seawater [Ca$^{2+}$] alone (Fig. 2c, 4b,c, Table
3b,c). Nevertheless, whereas the response to $\Omega_{Aragonite}$ was similar for both hybrid populations
they differed significantly in their response to [Ca$^{2+}$][HCO$_3^-$]/[H$^+$] (Table 4c,d).
3.3 Calculation of seawater [Ca$^{2+}$], $\Omega$ and [Ca$^{2+}$][HCO$_3^-$]/[H$^+$] for the Baltic Sea
Calculations of seawater [Ca$^{2+}$] were performed for the salinity range observed at the
collections sites of *M. edulis*-like and *trossulus*-like in Kiel Fjord and Usedom, respectively. In
Kiel Fjord, salinity fluctuated substantially between 10.5-24.7 g kg$^{-1}$ in the period 2005 – 2009
which resulted in simultaneous strong variations of seawater [Ca$^{2+}$] between 3.6 – 7.7 mM
with a mean of 5.6 mM (Table 1, Fig. 1d). In contrast, salinity in Usedom was lower with
mean salinity of 7.1 g kg$^{-1}$ and, in absolute numbers, more stable (3.4-9.1 g kg$^{-1}$, Table 1).
Thus, seawater [Ca$^{2+}$] in Usedom was ranging between 1.5 and 3.2 mM with an average of
2.7 mM (Table 1, Fig. 2d).
Calculation of [Ca$^{2+}$] along the Baltic salinity gradient revealed that the critical concentrations
of 3 and 2.5 mM at which calcification is negatively affected are reached at a salinity of about
7-8 g kg$^{-1}$, respectively, in all four sub regions (Fig. 5a). In contrast, calculated values for
[HCO$_3^-$]/[H$^+$] are above 0.13 in almost all regions within the distribution range of mussels as
long as the seawater is in equilibrium with current atmospheric CO$_2$ concentrations (Fig. 5b)
Only in the Gulf of Bothnia, critical values lower than 0.1 are observed for salinities of 4.5 g
kg$^{-1}$ and below. For $\Omega_{Aragonite}$, undersaturation is observed at a salinity of 9 g kg$^{-1}$ for the
central Baltic. The Gulfs of Bothnia and Finland are always undersaturated for $\Omega_{Aragonite}$, but
the Gulf of Riga seawater is supersaturated (Fig. 5c) and strong negative effects on larval
calcification can be expected for salinities of about 5 g kg$^{-1}$. Similarly, critical values for



$[Ca^{2+}][HCO_3^-]/[H^+]$ of 0.3 at which PD I formation is significantly affected are reached at a
salinity of 5 g kg$^{-1}$ in most regions of the Baltic excluding the Gulf of Riga (Fig. 5d).
Conditions for calcification will become more adverse in future as atmospheric $CO_2$
concentrations are going to reach 800 ppm. In this scenario, critical values for $[HCO_3^-]/[H^+]$
will be observed in most areas of Baltic at salinities below 10 g kg$^{-1}$ (Fig. 6b). In particular,
$[Ca^{2+}][HCO_3^-]/[H^+]$ and $\Omega_{Aragonite}$ will be below the critical threshold in all areas of the Baltic
Sea (Fig. 6c,d).
**4. Discussion**
This study investigated the impact of modifications of seawater $[Ca^{2+}]$ and carbonate
chemistry on shell formation of bivalve larvae. The experimental results were compared to
the environmental conditions prevailing in the Baltic Sea.
The laboratory experiments revealed that seawater $[Ca^{2+}]$ is a critical factor for shell
formation in marine bivalves. Similarly, $Ca^{2+}$ deposition into the shells of *Crassostrea gigas*
larvae following PD I formation was similar at seawater $[Ca^{2+}]$ of 10 and 16.8 mM but
reduced by 40% at 6.1 mM (Maeda-Martinez 1987). Thus, where high oceanic $[Ca^{2+}]$ of ~ 10
mM is not limiting bivalve calcification the low concentrations present in estuaries such as the
Baltic, significantly affect biomineralization.
In both tested populations, *M. edulis*-like and *M. trossulus*-like the overall response curve
was similar and both populations become calcium limited at $[Ca^{2+}]$ below 3 mM. *M. trossulus*-
like appeared to be slightly more tolerant to lowered $[Ca^{2+}]$ as larvae maintained larger PD I
lengths at similar $[Ca^{2+}]$ and PD I formation was successfully accomplished at 1.1 mM. The
response matches seawater $[Ca^{2+}]$ observed in the respective habitats of the tested
populations and may result from either phenotypic plasticity or genetic adaptation. It is also
possible that *M. edulis*-like living in the western brackish Baltic may have already adapted to
lower $[Ca^{2+}]$ compared to populations and species living in habitats characterized by higher
$[Ca^{2+}]$ (Maeda-Martinez 1987). As PD I formation is a crucial but sensitive stage during larval
life, impaired calcification by low $[Ca^{2+}]$ can have significant effects on larval performance
and fitness. As the distribution of bivalves is depending on successful larval dispersal, low
$[Ca^{2+}]$ can be an important factor which determines the distribution limits of mussels and
represents a strong selective force. Additionally, the strong $[Ca^{2+}]$ gradient observed between
the western Baltic-Kattegat transition zone and the central Baltic Sea can be one explanation
for the simultaneously observed allele frequency shift from *M. edulis*-like to *trossulus*-like
(Larsson et al. 2016, Stuckas et al. 2017).
Nevertheless, larval shell formation of Baltic mytilids starts to become $[Ca^{2+}]$ limited at
concentrations of about 3 mM and was significantly affected at 2.5 mM. Consequently, in
areas of the Baltic with salinities below 7-8 g kg$^{-1}$ and corresponding $[Ca^{2+}]$ < 3 mM, reduced
shell formation starts to compromise overall larval performance. At the critical salinity of 4.5 g
kg$^{-1}$ which delineates the distribution boundary of mussels in the Baltic (Westerbom et al.
2002), $[Ca^{2+}]$ is as low as 1.8 mM whereby concentration below 2 mM substantially impaired
PD I formation in our experiments. Importantly, even under these adverse conditions larvae
were viable and continued active swimming for up to 7 days. Thus impaired calcification in
low $[Ca^{2+}]$ seawater can result from two mechanisms acting independently or in combination:
I) continuous dissolution of existing calcium carbonate crystals under highly corrosive
conditions may prevent further net calcification or II) larvae only use a pre-determined
fraction of the energy stored in the egg for calcification. If this amount is not sufficient to
sustain full PD I formation under low $[Ca^{2+}]$ the budget does not seem to be adjusted to
provide additional energy to complete calcification. Instead larvae do not continue
calcification and may switch to an energy saving mode to stay alive. In our experiments, *M.
trossulus*-like apparently developed a higher tolerance to low $[Ca^{2+}]$ compared to *M. edulis*-
like but incipient impairment of calcification at about 3 mM was similar in both populations
which suggests relatively conserved $[Ca^{2+}]$ transport mechanisms in both populations.
Impact of external $[Ca^{2+}]$ on calcification has previously been studied mostly in corals for
which a significant correlation was observed in a number of studies (e.g. Chalker 1976; Ip
and Krishnaveni 1991). Whereas cytosolic calcium concentration are tightly regulated and





kept constantly low, calcifiers obviously developed a mechanism to accumulate high $[Ca^{2+}]$ in
specialized compartments within or outside the cell for biomineralization. In corals, $Ca^{2+}$
uptake and transport to the site of calcification is driven by a combination of diffusive and
active transport and involves active transport by plasma membrane $Ca^{2+}$-ATPase (PMCA,
Tambutte et al. 1996; Barott et al. 2015). In bivalves, calcification is performed by the outer
mantle epithelium (OME) or the shell field in adults and larvae, respectively (Kniprath 1980),
and a PMCA homolog has been localized in the OME of oysters and its inhibition negatively
impacted shell growth in freshwater clams which might suggest a conserved function in
bivalve calcification as well (Wang et al. 2008; Zhao et al. 2016).
Early studies suggested that the extrapallial fluid (EPF) of bivalves provides the microhabitat
for calcification (Crenshaw 1972). However, $[Ca^{2+}]$ and acid-base status of bulk EPF of adult
mussels corresponds to seawater and haemolymph conditions, respectively, which supports
excretion of $CO_2$ via passive diffusion into the ambient seawater (Thomsen et al. 2010;
Heinemann et al. 2012). In *M. edulis*-like larvae, kept above the critical threshold of 3 mM,
CS $[Ca^{2+}]$ was marginally but significantly elevated compared to seawater $[Ca^{2+}]$. At lowered
environmental $[Ca^{2+}]$ between 2-3 mM CS $[Ca^{2+}]$ was not significantly enriched compared to
seawater concentration. At these seawater $[Ca^{2+}]$, calcification rates were significantly
reduced but larvae were still able to produce a smaller but complete PD I. At even lower
ambient $[Ca^{2+}]$ of 1.5 mM, CS $[Ca^{2+}]$ was again significantly elevated compared to seawater
which was, however, accompanied by strongly reduced PD I formation. The incapacity of
larvae to maintain transmembrane $Ca^{2+}$ transport at lowered $[Ca^{2+}]$ potentially indicates a
significant contribution of diffusion or involvement of a low affinity $Ca^{2+}$ transporter (e.g.
$Na^+/Ca^{2+}$ Exchanger) in this process (Blaustein and Lederer 1999). Thus, larvae may actively
enrich CS $[Ca^{2+}]$ to increase $\Omega_{Aragonite}$ and support the structural integrity of the shell under
corrosive conditions. Alternatively, CS $[Ca^{2+}]$ only increased secondarily as a result of
drastically reduced calcification rates.
In the present study, the effect of lowered $[Ca^{2+}]$ was most pronounced under conditions
when seawater carbonate chemistry was not a limiting parameter for calcification. Lowering
of seawater $C_T$, which has a similar effect on $\Omega_{Aragonite}$ and $[HCO_3^-]/[H^+]$ as acidification,
significantly affects the rate of PD I formation. Under these $C_T$ / $HCO_3^-$ limiting conditions,
seawater $[Ca^{2+}]$ had only a minor, yet slightly positive, linear effect on shell formation.
Presumably the effect was smaller as $Ca^{2+}$ uptake was not any longer the only rate limiting
process but rather $HCO_3^-$ uptake and / or $H^+$ extrusion (Bach 2015) or impaired kinetics of
crystal formation (Waldbusser et al. 2014).
Importantly, the applied experimental seawater manipulations of calcium and carbonate
chemistry can be integrated by calculation of $\Omega_{Aragonite}$ or extending the SIR term to
$[Ca^{2+}][HCO_3^-]/[H^+]$ which also takes lowered availability of $[Ca^{2+}]$ into account (Bach 2015;
Fassbender et al. 2016). Plotting shell length against these two parameters revealed a
similar response for all manipulations independent whether they were manipulated by
lowered $[Ca^{2+}]$ or $C_T$. The correlation of calcification with these parameters corresponded to
previously observed shell formation performance of mussels and oysters resulting from
modifications of seawater carbonate chemistry only (Waldbusser et al. 2014; Waldbusser et
al. 2015; Thomsen et al. 2015). As salinity and temperature were not changed in the
experiments performed with *M. edulis*-like $\Omega_{Aragonite}$ and $[Ca^{2+}][HCO_3^-]/[H^+]$ are linearly
correlated and it is not possible to distinguish whether shell formation is modified by the
changed kinetics of crystal formation (Waldbusser et al. 2015), higher dissolution due to
undersaturation of the EPF with respect to calcium carbonate (Miller et al. 2009; Thomsen et
al. 2010; Melzner et al. 2011, Frieder et al. 2017) or by lowered substrate availability and
impaired $H^+$ removal from the calcifying fluids (Thomsen et al. 2015; Bach 2015; Fassbender
et al. 2016). However, the calcification response of *M. trossulus*-like was similar to *M. edulis*-
like when plotted against $\Omega_{Aragonite}$ but differed significantly for $[Ca^{2+}][HCO_3^-]/[H^+]$ in
accordance with the higher tolerance to lowered $[Ca^{2+}]$. This could indicate local adaptation
of *M. trossulus*-like to the adverse environment in the low saline areas of the Baltic. In
contrast, the response to $\Omega_{Aragonite}$ was similar in animals from both populations which may



indicate that shell dissolution under corrosive conditions impacts net shell formation to the
same extent.
Our experimental data revealed that larval calcification is substantially compromised by
environmental conditions encountered in the Baltic Sea. Calculation of Baltic seawater $[Ca^{2+}]$
suggests $[Ca^{2+}]$ limitation of calcification at salinities of about 8 g $kg^{-1}$. Thus, with exception of
the western Baltic Sea with its higher salinity values, mussels inhabiting most areas of the
Baltic suffer from low $Ca^{2+}$ availability. Interestingly, studies measuring Baltic Sea $[Ca^{2+}]$
revealed increasing concentrations over the last decades which may have a beneficial effect
on calcification for a given salinity (Kremling and Wilhelm 1997). Nevertheless, the expected
overall reduction of salinity will most likely exceed the minor positive effect of $[Ca^{2+}]$
enrichment and negatively affect overall fitness   by osmotic stress and secondarily
calcification (Gräwe et al. 2013).
In contrast to $[Ca^{2+}]$, estimating current carbonate chemistry for the four Baltic sub regions
suggests that the influence is of less importance for limitation of calcification. The calculated
$[HCO_3^-]/[H^+]$ and $\Omega_{Aragonite}$ for seawater in equilibrium with current atmospheric $CO_2$
concentrations remain above the critical thresholds of 0.1-0.13 and 1, respectively (Thomsen
et al. 2015, this study). However, this conclusion does not consider the substantial variability
of carbonate chemistry in the surface water of the Baltic which is modified by biogeochemical
processes such as riverine composition, photosynthesis and upwelling on a seasonal and
spatial scale. Seawater carbonate chemistry can be substantially modified by phytoplankton
blooms in spring and early summer causing a draw down of seawater $pCO_2$ to 150 µatm
thereby causing elevated pH, $[CO_3^{2-}]$ and $[HCO_3^-]/[H^+]$ for several weeks (Schneider and
Kuss 2004). Consequently, larvae can be exposed to environmental conditions which are
beneficial for calcification. In contrast, local upwelling phenomena have the opposite effect
leading to lowered pH and $[CO_3^{2-}]$, $[HCO_3^-]/[H^+]$ and elevated $pCO_2$ (Thomsen et al. 2010;
Saderne et al. 2013). Upwelling events are common in the Baltic Sea in particular along the
western coastlines (Myrberg and Andrejev 2003). However, research mostly focused on the
effect of upwelling on temperature and nutrient supply but neglected the local impacts on
carbonate chemistry (e.g. Haapala 1994). As upwelling causes rapid elevation of $pCO_2$ within
a short period of hours but can last for several days to few weeks, thus for a significant part
of a larval life time, its impact on calcification and performance of larvae can be substantial
(Barton et al. 2012; Thomsen et al. 2015, 2017).
In addition to the present carbonate system variability, the successive increase of
atmospheric $CO_2$ concentrations and coupled pH decline in the Baltic will result in
progressively adverse conditions for calcification. This process is particularly critical for
mussel populations inhabiting the low saline areas of the Baltic where conditions for
calcification are less favourable already today and will become more adverse in the future.
Nevertheless, it has recently been shown that increasing $A_T$ (from an unaccounted source)
may partly and even completely compensate the negative effects of $CO_2$ uptake (Müller et al.
2016). Consequently, bivalve calcification may benefit from higher $A_T$ and thus favourable
carbonate chemistry in future, but lowered salinity might still affect performance.
Both substrates relevant for calcification, $Ca^{2+}$ and inorganic carbon are integrated in the
terms $\Omega$ and the SIR extended to $[Ca^{2+}][HCO_3^-]/[H^+]$. In fact the calcification response of
bivalve larvae in our experiments was accurately described by both terms for a given salinity
and temperature. Nevertheless, calculations of the environmental conditions in the four Baltic
sub regions revealed important differences. $\Omega_{Aragonite}$ remains favourable for calcification (>1)
in most parts of the central Baltic and in the Gulf of Riga caused by high alkaline riverine
runoff and therefore prohibits dissolution of shell crystals (Juhna and Klavins 2000). In
contrast, calculated values for $[Ca^{2+}][HCO_3^-]/[H^+]$ are below the critical threshold of 0.7 in all
sub regions at a salinity of 11 g $kg^{-1}$ caused by low $[Ca^{2+}]$. Thus, it is of high ecological
relevance whether bivalve calcification is sensitive to the reduced kinetic of shell formation
and dissolution depending on $\Omega$ or lowered substrate availability and inhibition by $[H^+]$.
According to our experimental data most likely a combination of both parameters is
determining sensitivity. However, compared to *M. edulis*-like, *M. trossulus*-like seems to have
evolved a slightly higher tolerance to low $[Ca^{2+}][HCO_3^-]/[H^+]$, but not to low $\Omega_{Aragonite}$. A similar





response has been observed in a comparison between Baltic and North Sea mussels under
simulated ocean acidification (Thomsen et al. 2017).
In conclusion, this study reveals strong impacts of lowered [$Ca^{2+}$] and carbonate chemistry,
which are naturally changing along the Baltic salinity gradient, on the early calcification of
mussel larvae. Strong delays and impairment of complete shell formation most likely affect
the energy budget and overall physiology of mussels in the low saline areas. Consequently,
low [$Ca^{2+}$] and adverse carbonate chemistry impact mussel fitness substantially and
therefore likely seem to contribute significantly in determining the distribution of marine
mussels in estuaries such as the Baltic Sea.
**Author Contributions:**
JT conceived the study and led the writing of the manuscript; JT, KR, TS and FM collected
data; JT, KR, MB, and FM analysed the data. All authors contributed to the various
manuscript drafts.
**Acknowledgements:**
The authors thank Thomas Stegmann for performing $Ca^{2+}$ measurements, Marian Hu for
supporting $Ca^{2+}$-microelectrode measurements and Ulrike Panknin for maintaining
*Rhodomonas* cultures. Furthermore, Detlev Machoczek and Rainer Kiko are acknowledged
for providing and supporting processing of Oder Bank salinity data, respectively. This study
was funded by the BMBF program BIOACID subproject 2.3 and CACHE, a Marie Curie Initial
Training Network (ITN) funded by the People Programme (Marie Curie Actions) of the
European Union's Seventh Framework Programme FP7/2007-2013/ under REA grant
agreement n°[605051]13. The authors declare no conflict of interest.
**Data availability:**
All data are available under: Thomsen, Jörn; Ramesh, Kirti; Sanders, Trystan; Bleich, Markus;
Melzner, Frank (2017): Effects of seawater calcium on calcification in mussel larvae.
*PANGAEA*, Unpublished dataset #871804.

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



Fig. 1 Bathymetric map of the Baltic Sea and its sub regions which are characterized by
specific carbonate chemistry. Sampling spots for mussel populations used in the experiments
are indicated by light blue dots.

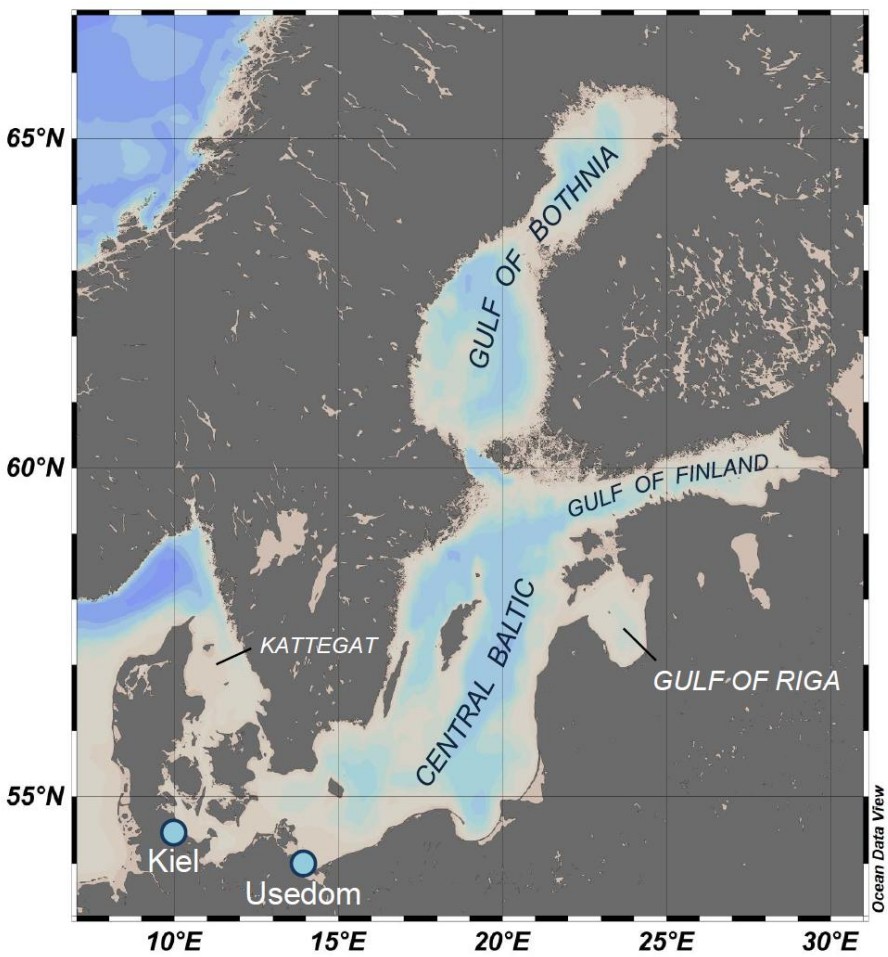






Fig. 2 Prodissoconch I length of mussel larvae as a function of seawater [Ca$^{2+}$]. A) *M. edulis*-
like, different symbols represent different experimental runs (1-5) B) *M. trossulus*-like, C)
Comparison of *M. edulis*-like and *trossulus*-like, D) Boxplots of seawater [Ca$^{2+}$] at the
collection site in Kiel Fjord and at Usedom depicting median, 25 and 75% quartiles and
outliers.





Fig. 3 [$Ca^{2+}$] in the calcifying space (CS) of *M. edulis*-like larvae. A) CS [$Ca^{2+}$] as a function of seawater [$Ca^{2+}$], the line indicates the isoline B) Difference between CS [$Ca^{2+}$] and seawater [$Ca^{2+}$] at four [$Ca^{2+}$] treatments expressed as [$Ca^{2+}$]$_{CS}$-[$Ca^{2+}$]$_{SW}$. Bar chart depicts mean ± standard error of the mean (N=6).

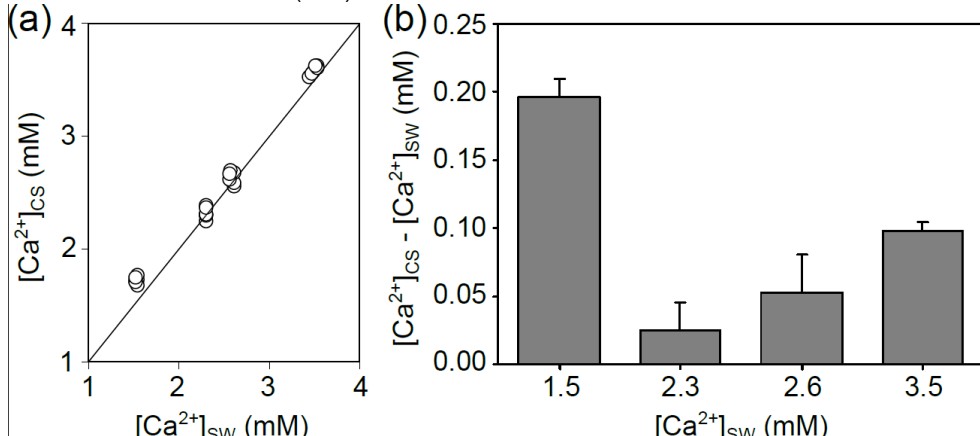



Fig. 4 Prodissoconch I length of mussel larvae exposed to varying $C_T$ and $[Ca^{2+}]$ plotted
against A) $[Ca^{2+}]$, B) $\Omega_{Aragonite}$, C) $[Ca^{2+}][HCO_3^-]/[H^+]$.

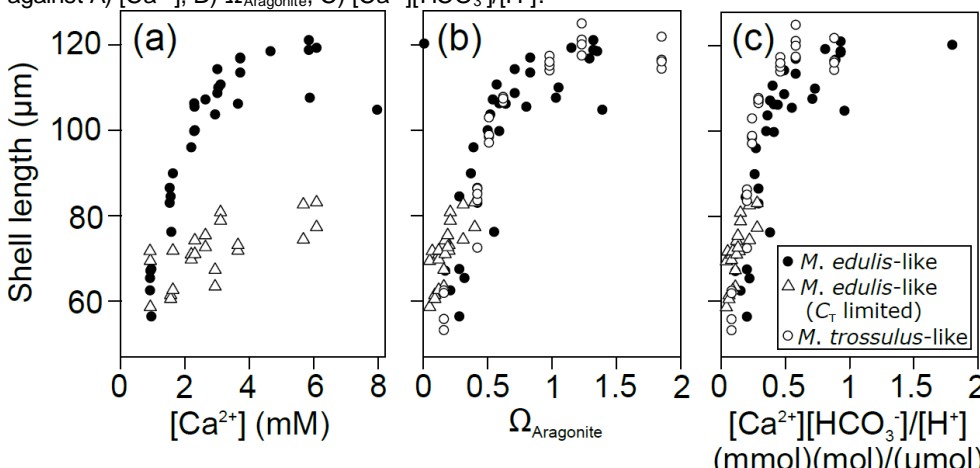






Fig. 5 Environmental parameters relevant for calcification in the Baltic Sea calculated for
current salinity-$A_T$ correlations and atmospheric $CO_2$ concentration (400 ppm). A) [$Ca^{2+}$], B)
[$HCO_3^-$]/[$H^+$], C) $\Omega_{Aragonite}$ and D) [$Ca^{2+}$][$HCO_3^-$]/[$H^+$] plotted against salinity for the four sub
regions of the Baltic Sea. Dashed lines and grey areas indicate conditions of incipient and
significant reduction of larval calcification rates, respectively.

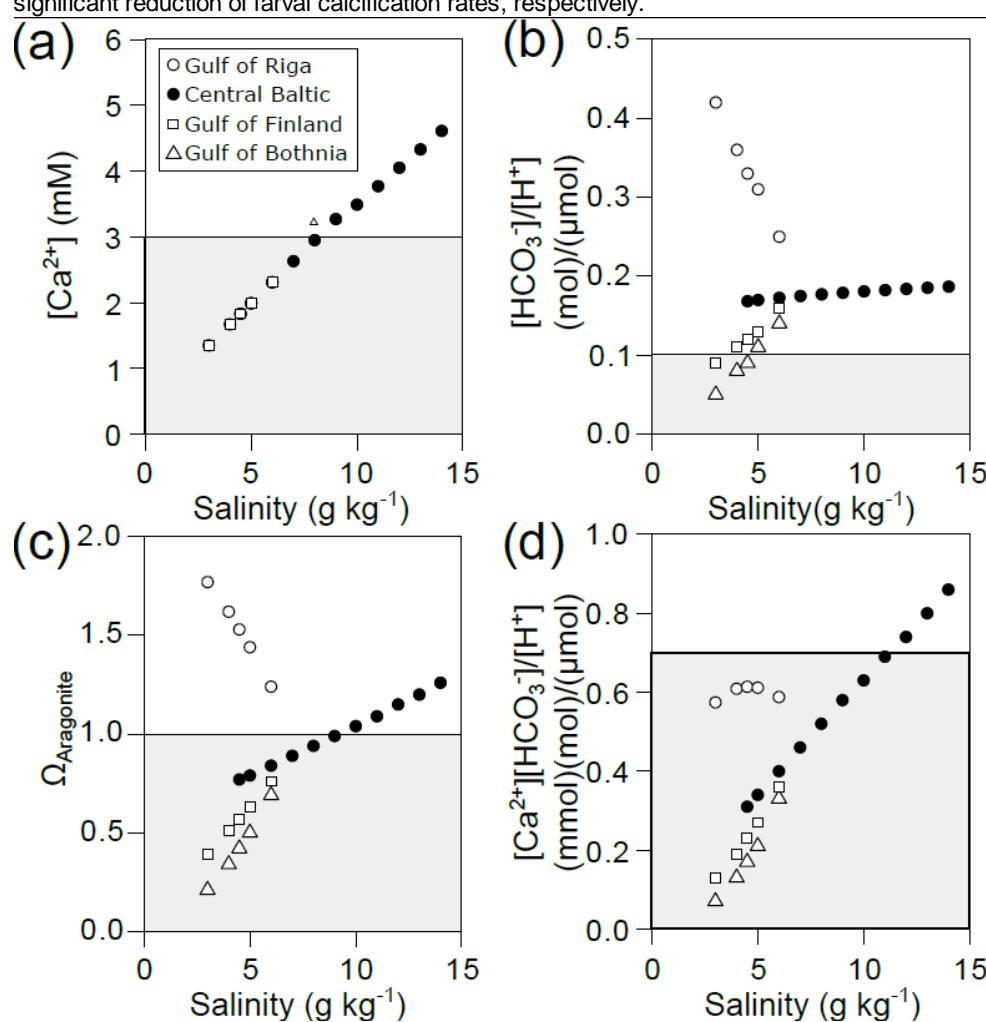






Fig. 6 Predicted environmental parameters relevant for calcification in the Baltic Sea
calculated for current salinity-$A_T$ correlations and future atmospheric $CO_2$ concentration (800
ppm). A) $[Ca^{2+}]$, B) $[HCO_3^-]/[H^+]$, C) $\Omega_{Aragonite}$ and D) $[Ca^{2+}][HCO_3^-]/[H^+]$ plotted against salinity
for the four sub regions of the Baltic Sea. Dashed lines and grey areas indicate conditions of
incipient and significant reduction of larval calcification rates, respectively.

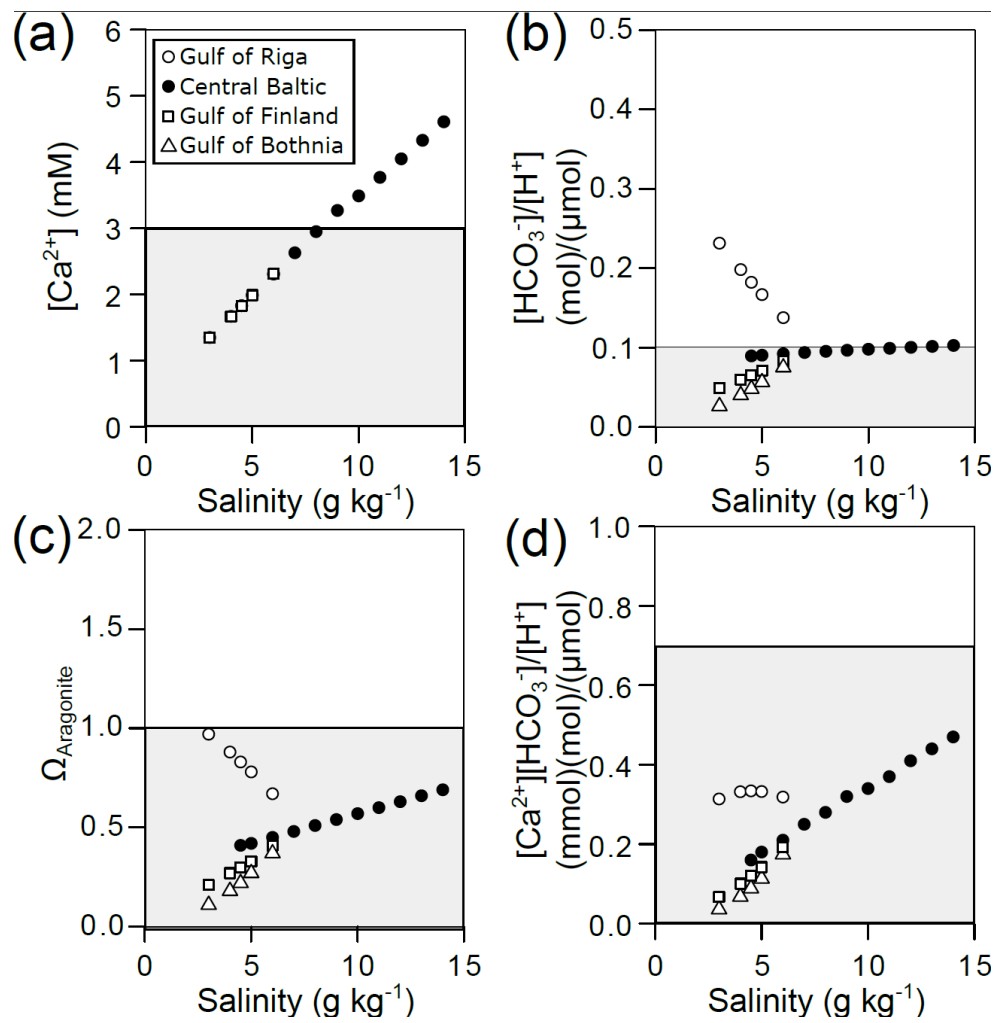






Table 1. Natural variability of salinity and [Ca$^{2+}$] in Kiel Fjord and Usedom.

| Salinity (g kg$^{-1}$) | Usedom | Kiel |
|---|---|---|
| Min. | 3.44 | 10.50 |
| 1st Qu. | 6.81 | 15.30 |
| Median | 7.19 | 17.10 |
| Mean | 7.14 | 17.15 |
| 3rd Qu. | 7.74 | 18.90 |
| Max. | 9.33 | 24.70 |

| [Ca$^{2+}$] (mM) | Usedom | Kiel |
|---|---|---|
| Min. | 2.22 | 3.57 |
| 1st Qu. | 2.67 | 4.97 |
| Median | 2.71 | 5.49 |
| Mean | 2.70 | 5.51 |
| 3rd Qu. | 2.75 | 6.01 |
| Max. | 3.14 | 7.70 |




Table 2: Experimental conditions during larval experiments, N:1-10 determinations, $\Omega_{Aragonite}$
and $[Ca^{2+}][HCO_3^-]/[H^+]$ are calculated from measured $[Ca^{2+}]$, $C_T$ and $pH_{NBS}$.

A) $[Ca^{2+}]$ manipulation experiments with *M. edulis*-like

| $[Ca^{2+}]$ treatment | $[Ca^{2+}]$ (mmol/L) |
|---|---|
| <1 mM | 0.86 ± 0.02 |
| 1.5 - 2 mM | 1.56 ± 0.03 |
| 2.0 - 2.5 mM | 2.19 ± 0.03 |
| 2.5 - 3 mM | 2.82 ± 0.05 |
| 3.0 - 4.0 mM | 3.62 ± 0.06 |
| 4.0 - 5.0 mM | 4.42 ± 0.11 |
| 5.0 - 6.0 mM | 5.74 ± 0.07 |
| 6.0 - 8.0 mM | 6.83 ± 0.25 |
| >8.0 mM | 9.22 ± 0.10 |

B) $[Ca^{2+}]$ manipulation experiments with *M. trossulus*-like

| $[Ca^{2+}]$ treatment | $[Ca^{2+}]$ (mmol/L) | $\Omega_{Aragonite}$ | $[Ca^{2+}][HCO_3^-]/[H^+]$ [mmol][mol]/[μmol] |
|---|---|---|---|
| <1 mM | 0.40 ± 0.02 | 0.16 ± 0.02 | 0.08 ± 0.01 |
| 1 mM | 1.07 ± 0.04 | 0.43 ± 0.00 | 0.20 ± 0.01 |
| 1-1.5 mM | 1.36 ± 0.00 | 0.51 ± 0.03 | 0.24 ± 0.01 |
| 1.5 - 2 mM | 1.79 ± 0.03 | 0.62 ± 0.04 | 0.29 ± 0.02 |
| 2.5 - 3 mM | 2.94 ± 0.03 | 0.98 ± 0.07 | 0.46 ± 0.03 |
| 3.0 - 4.0 mM | 3.74 ± 0.04 | 1.23 ± 0.06 | 0.58 ± 0.03 |
| >5.0 mM | 5.78 ± 0.01 | 1.86 ± 0.11 | 0.88 ± 0.04 |

C) $[Ca^{2+}]$ and carbonate systems manipulation experiments with *M. edulis*-like

| treatment | $[Ca^{2+}]$ (mmol/L) | $\Omega_{Aragonite}$ | $[Ca^{2+}][HCO_3^-]/[H^+]$ [mmol][mol]/[μmol] |
|---|---|---|---|
| control + high $C_T$ | 0.93 ± 0.02 | 0.26 ± 0.07 | 0.18 ± 0.05 |
| | 1.55 ± 0.03 | 0.45 ± 0.09 | 0.31 ± 0.06 |
| | 2.25 ± 0.06 | 0.64 ± 0.15 | 0.44 ± 0.10 |
| | 2.99 ± 0.05 | 0.80 ± 0.22 | 0.55 ± 0.15 |
| | 3.69 ± 0.04 | 1.05 ± 0.23 | 0.73 ± 0.16 |
| | 5.45 ± 0.70 | 1.36 ± 0.04 | 0.94 ± 0.02 |
| | 8.69 ± 1.03 | 2.63 | 1.8 |
| low $C_T$ | 0.92 ± 0.01 | 0.06 ± 0.01 | 0.04 ± 0.01 |
| | 1.59 ± 0.05 | 0.10 ± 0.03 | 0.07 ± 0.02 |
| | 2.25 ± 0.08 | 0.14 ± 0.03 | 0.10 ± 0.02 |
| | 2.78 ± 0.21 | 0.17 ± 0.02 | 0.12 ± 0.01 |
| | 3.37 ± 0.38 | 0.20 ± 0.01 | 0.14 ± 0.01 |
| | 5.88 ± 0.29 | 0. 36 ± 0.06 | 0.25 ± 0.05 |






Table 3: Model parameters (a, b, c) describing PD I size as a function of experimental
seawater conditions for *Mytilus edulis*-like and *trossulus*-like: Shell length (µm) = a+ b *
e^(c*[parameter]).

A) Seawater $[Ca^{2+}]$

| *M. edulis*-like | Estimate | std Error | t-value | p |
|---|---|---|---|---|
| a | 112.7 | 1.8 | 63.4 | <0.001 |
| b | -100.7 | 7.6 | -13.3 | <0.001 |
| c | -0.8 | 0.1 | -9.3 | <0.001 |
| *M. trossulus*-like | Estimate | std Error | t-value | p |
| a | 120.6 | 1.8 | 66 | <0.001 |
| b | -94.5 | 5.2 | -18.1 | <0.001 |
| c | -1 | 0.1 | -10.3 | <0.001 |

B) Seawater $\Omega_{Aragonite}$

| *M. edulis*-like | Estimate | std Error | t-value | p |
|---|---|---|---|---|
| a | 118.9 | 3.8 | 31.1 | <0.001 |
| b | -106.1 | 16.1 | -6.6 | <0.001 |
| c | -3.1 | 0.6 | -4.7 | <0.001 |
| *M. trossulus*-like | Estimate | std Error | t-value | p |
| a | 121.6 | 2.3 | 53.5 | <0.001 |
| b | -100.8 | 6.4 | -15.7 | <0.001 |
| c | -2.8 | 0.3 | -9.0 | <0.001 |

C) Seawater $[Ca^{2+}][HCO_3^-]/[H^+]$

| *M. edulis*-like | Estimate | std Error | t-value | p |
|---|---|---|---|---|
| a | 125.9 | 5.0 | 25.3 | <0.001 |
| b | -73.5 | 4.3 | -17.2 | <0.001 |
| c | -1.8 | 0.3 | -5.9 | <0.001 |
| *M. trossulus*-like | Estimate | std Error | t-value | p |
| a | 121.4 | 2.2 | 54.0 | <0.001 |
| b | -104.8 | 7.1 | -14.9 | <0.001 |
| c | -6.0 | 0.7 | -9.0 | <0.001 |






Table 4: Results for linear models fitted on log transformed data of shell length and seawater
parameters, significant results in bold.

A) Response to $[Ca^{2+}]$

|  | Estimate | std Error | t-value | p |
|---|---|---|---|---|
| **Intercept** | **4.17** | **0.07** | **59.2** | **<0.001** |
| **$Ca^{2+}$** | **0.31** | **0.06** | **4.9** | **<0.001** |
| **population** | **0.12** | **0.04** | **2.8** | **<0.01** |
| $Ca^{2+}$:population | -0.01 | 0.04 | -0.3 | >0.05 |
| F: 82.1 | p: <0.001 | R2: 0.77 | | |

B) Response to $\Omega_{Aragonite}$

|  | Estimate | std Error | t-value | p |
|---|---|---|---|---|
| **Intercept** | **4.64** | **0.05** | **90.4** | **<0.001** |
| **$\Omega_{Aragonite}$** | **0.13** | **0.04** | **3.08** | **<0.01** |
| population | 0.04 | 0.03 | 1.23 | >0.05 |
| **$\Omega_{Aragonite}$: population** | **0.1** | **0.03** | **2.86** | **<0.01** |
| F: 116.4 | p:<0.001 | R2: 0.82 | | |

C) Response to $[Ca^{2+}][HCO_3^-]/[H^+]$ (CHH)

|  | Estimate | std Error | t-value | p |
|---|---|---|---|---|
| **Intercept** | **4.69** | **0.08** | **60.1** | **<0.001** |
| **CHH** | **0.27** | **0.07** | **3.8** | **<0.001** |
| **population** | **0.13** | **0.05** | **2.5** | **<0.05** |
| CHH: population | 0.02 | 0.04 | 0.5 | >0.05 |
| F: 67.4 | p: <0.001 | R2: 0.78 | | |
