# Peer review of "Calcification in a marginal sea – influence of seawater [Ca2+] and carbonate chemistry on bivalve shell formation"

_Biogeosciences, 2017_

## Referee Comment (RC1) · Anonymous Referee #1 · 8 Nov 2017

This manuscript deals with the effects of concentrations of calcium and dissolved inorganic carbon species on early shell developments in the mussels. The formation of the prodissoconch I are considered in terms of environmental conditions and, importantly in the context of ontogeny. The strength of the manuscript is a precise experimental design, that it deals with this biological process in detail in the environmental, onogenetic and genetic contexts, providing important insight for ocean acidification. The authors present detailed discussion on this phenomenon by way of the chemistry of calcification space indicating thresholds of the calcification responses to carbonate undersaturation. Thomsen et al. provide important data, serving to motivate more detailed future experiments/monitoring studies.

[Figure]

Some points: The critical [Ca2+] and saturation thresholds are well characterized from the larval shell length. Is shell morphology likely to relate to tolerance to lowered [Ca2+] and DIC in prodissoconch I? Please provide the relationship between environmental factors, the number and the presence of abnormal individuals.

Line 70: Is there any similar environmental influence on the formation of prodissoconch II? Please provide some information.

The variation of seawater Mg/Ca are also known to have influence on marine biological calcification. Please discuss about the potential impact of varying seawater [Mg2+]/[Ca2+] of this experiments on bivalve shell formation. Suggested reading: Ries, J.B. (2010) Review: geological and experimental evidence for secular variation in seawater Mg/Ca (calcite-aragonite seas) and its effects on marine biological calcification, Biogeosciences, 7, 2795–2849.

Upper- and lower-cases in captions and figures should be unified (e.g., 2A, 2a)

―――――――――――――――――――――――――

---

## Referee Comment (RC2) · Anonymous Referee #2 · 8 Nov 2017

General comments:

This study addresses the impact of calcium ion concentration on first shell formation in bivalve larvae (Mytilid mussels) independent of salinity and in concert with changes in seawater carbonate chemistry associated with ocean acidification. The impacts of changing seawater chemistry on the biology of marine calcifiers has been the topic of extensive study in the last decade particularly with respect to changes in pH and carbonate chemistry under current and projected changes in atmospheric CO2 concentrations. The current study addresses a potential compounding factor-low levels of calcium ions for calcification and shell formation in brackish waters of the Baltic Sea.

[Figure]

The approach employed combines 1) evaluation of biological response (capacity for first shell formation, ion concentrations in the extracellular calcification space) under experimentally manipulated calcium ion concentrations alone, and in connection with altered carbonate chemistry and 2) a comparison of the experimental findings with environmental conditions (and variation) in the Baltic.

The experimental approach was appropriate and well executed. I appreciate the efforts to address biological responses at the level of the whole organism as well as components of physiological response (ion concentrations). Further, the synthesis of environmental data contextualizes the biological responses for discussion of their implications.

I have included below several questions and comments that may guide the authors in refining and clarifying their presentation of the study.

Specific comments:

-Is shell length along a sufficient assessment of the impact of seawater chemistry on calcification? Using shell length as the metric for calcification assumes that a relationship between length and mass of the shell is consistent over different environmental treatments and there are indications this is not the case (Frieder et al 2017 ICES JMS, Gaylord et al 2011 JEB).

-It would be helpful to clarify how "larvae that had not developed a complete PD I shell" were assessed. Does this mean shell was abnormal? Partially developed? Would they have developed given more time, or a change in conditions? For instance, at Line 263, were the 7 day old larvae with a 63.7 $\mu$m shell diameter just small versions of a complete PD I?

-As presented, I don't find the suggestion that the "troussulus-like" animals have evolved higher tolerance for low calcium any more compelling than the possibility that they are acclimatized to the prevailing conditions experienced at the collection site. The

suggestion is of course interesting based upon East-West gradient of allele frequencies from Stuckas et al. However, in the absence of any data to validate that the broodstock collected from the different sampling sites are genetically distinct adaptation doesn't seem to be more or less favored over a plasticity argument.

-Can any comment be included on the historical distribution of these populations suggesting that they have expanded/contracted?

-I am having a hard time interpreting figure 3B. CS calcium continues to decline with the reduction of seawater calcium, but it is no longer following seawater. You propose that this is a surplus of calcium stored up when shell formation rate plummets? So, some other component of the calcifying process is inhibited at this level, and must be limiting the utilization of this calcium?

Minor comments/corrections:

-Line 120-"Finish" should be "Finnish"

-Fig 3 is introduced in the results before Fig 2b,c,d. Further, the paragraph (starting at line 271) regarding results of microelectrode measurements should probably be the last the paragraph of section 3.1. This would move the current, last paragraph of the section (which contains reference to Fig 2bcd) into the appropriate presentation order.

-Line 280-reference to Figure 1, should be Figure 2

-It would be helpful to indicate the control treatment levels on Fig 2 since higher and lower [Ca2+] treatments are applied

---

## Author Comment (AC1) · 5 Dec 2017

This manuscript deals with the effects of concentrations of calcium and dissolved inorganic carbon species on early shell developments in the mussels. The formation of the prodissoconch I are considered in terms of environmental conditions and, importantly in the context of ontogeny. The strength of the manuscript is a precise experimental design, that it deals with this biological process in detail in the environmental, onogenetic and genetic contexts, providing important insight for ocean acidification. The authors present detailed discussion on this phenomenon by way of the chemistry of calcification space indicating thresholds of the calcification responses to carbonate undersatu-

ration. Thomsen et al. provide important data, serving to motivate more detailed future experiments/monitoring studies.

Response: We like to thank referee 1 for her/his constructive comments on our manuscript, please find our responses below.

Some points: The critical [Ca2+] and saturation thresholds are well characterized from the larval shell length. Is shell morphology likely to relate to tolerance to lowered [Ca2+] and DIC in prodissoconch I? Please provide the relationship between environmental factors, the number and the presence of abnormal individuals.

Response: Malformations are directly related to the inability to form a normal sized shell (ca. 110 $\mu$m) thus we only measured shell length but did not quantify numbers of malformed individuals. At shell a size below about 85 $\mu$m malformations such as a protruded mantle were observed, similar to the observations reported by the studies of His and co-workers. However, at moderately lowered Ca2+ concentrations this 'malformation' most likely corresponds to a delay of calcification as normal D-shells were observed when larvae continued growth for one more day. Only at Ca2+ concentrations <2 mM larvae did not complete D-shell formation and shells remained malformed/incomplete even after one week.

Line 70: Is there any similar environmental influence on the formation of prodissoconch II? Please provide some information.

Response: So far no longer studies have been run, but as calcification would still be limited by Ca2+ availability the outcome would be similar, as we also observed for settled juvenile mussels >1 mm(Sanders et al. in prep). The effect, however, is most prominent during PD I formation as calcification rates are much higher in this phase compared to later larval and juvenile life stages (Waldbusser et al .2014, Thomsen et al. 2015).

The variation of seawater Mg/Ca are also known to have influence on marine biological calcification. Please discuss about the potential impact of varying seawater [Mg2+]/[Ca2+] of this experiments on bivalve shell formation. Suggested reading: Ries, J.B. (2010) Review: geological and experimental evidence for secular variation in seawater Mg/Ca (calcite-aragonite seas) and its effects on marine biological calcification, Biogeosciences, 7, 2795–2849.

Response: We also performed experiments applying Mg2+ manipulations but did not observe a significant effect on PD I formation rates within the tested range. This does not exclude any effects on shell composition which was not tested in this study. We have added a section on this topic to the revised MS.

Upper- and lower-cases in captions and figures should be unified (e.g., 2A, 2a)

Response: corrected

––––––––––––––––––––––––––––

---

## Author Comment (AC2) · 5 Dec 2017

General comments: This study addresses the impact of calcium ion concentration on first shell formation in bivalve larvae (Mytilid mussels) independent of salinity and in concert with changes in seawater carbonate chemistry associated with ocean acidification. The impacts of changing seawater chemistry on the biology of marine calcifiers has been the topic of extensive study in the last decade particularly with respect to changes in pH and carbonate chemistry under current and projected changes in atmospheric CO2 concentrations. The current study addresses a potential compounding factor-low levels of calcium ions for calcification and shell formation in brackish waters of the Baltic Sea. The approach employed combines 1) evaluation of biological response (capacity for first shell formation, ion concentrations in the extracellular calcification space) under experimentally manipulated calcium ion concentrations alone, and in connection with altered carbonate chemistry and 2) a comparison of the experimental findings with environmental conditions (and variation) in the Baltic. The experimental approach was appropriate and well executed. I appreciate the efforts to address biological responses at the level of the whole organism as well as components of physiological response (ion concentrations). Further, the synthesis of environmental data contextualizes the biological responses for discussion of their implications. I have included below several questions and comments that may guide the authors in refining and clarifying their presentation of the study.

Response: We like to thank referee 2 for her/his constructive comments on our manuscript, please find our responses below.

Specific comments: -Is shell length along a sufficient assessment of the impact of seawater chemistry on calcification? Using shell length as the metric for calcification assumes that a relationship between length and mass of the shell is consistent over different environmental treatments and there are indications this is not the case (Frieder et al 2017 ICES JMS, Gaylord et al 2011 JEB).

Response: The study by Frieder et al. is one of the few which directly relate shell length and mass and according to the fig 7b both are linearly correlated. They responded similarly to adverse carbonate chemistry fig 1 a+b indicating a general impairment of calcification; therefore we assume a similar response for Ca2+ manipulation. Gaylord et al. did not used same sized animals for their measurements of shell thickness, thus did not prove a shift of the length mass correlation or thinning under adverse conditions.

-It would be helpful to clarify how "larvae that had not developed a complete PD I shell" were assessed. Does this mean shell was abnormal? Partially developed? Would they have developed given more time, or a change in conditions? For instance, at Line

263, were the 7 day old larvae with a 63.7 _m shell diameter just small versions of a complete PD I?

Response: The complete PD I shell of bivalve larvae have a clear D shape and covers the whole animal. Thus, animals which were not completely covered by the shell or those whose shell shape was still round were considered as 'abnormal'. In general, this only indicates a delay of calcification and the full shell was formed later. However, in treatments <2 mM Ca2+ the shell of Kiel Fjord specimens were not completed within 7 day and growth ceased. This suggests that under these conditions calcification was not simply delayed and but effectively affected. In the revised MS we will make this clearer.

-As presented, I don't find the suggestion that the "troussulus-like" animals have evolved higher tolerance for low calcium any more compelling than the possibility that they are acclimatized to the prevailing conditions experienced at the collection site. The suggestion is of course interesting based upon East-West gradient of allele frequencies from Stuckas et al. However, in the absence of any data to validate that the broodstock collected from the different sampling sites are genetically distinct adaptation doesn't seem to be more or less favored over a plasticity argument.

Response: The reviewer is correct, we cannot not rule out that plasticity has an influence on shell formation as well as we already observed during transgenerational exposure of the Kiel Fjord population to elevated pCO2 (Thomsen et al. 2017). However, Stuckas et al. (2017) documented that the gene exchange between both populations is strongly limited as larval drift does not allow for direct exchange. Thus, the stability of the genetic gradient supports that isolation of the populations is likely. In the revised version of MS we will discuss both possibilities.

-Can any comment be included on the historical distribution of these populations suggesting that they have expanded/contracted?

Response: The distribution of the population is most likely correlated with Baltic Sea

salinity and it changes over time. However, the lack of historical data does not allow to document the exact genetic cline within the Baltic.

-I am having a hard time interpreting figure 3B. CS calcium continues to decline with the reduction of seawater calcium, but it is no longer following seawater. You propose that this is a surplus of calcium stored up when shell formation rate plummets? So, some other component of the calcifying process is inhibited at this level, and must be limiting the utilization of this calcium?

Response: Figure 3a shows the decline of CS $Ca_{2+}$ with seawater $Ca_{2+}$ concentration. This decline continued in all treatments, but the CS concentration significantly deviated from SW in the lowest measured $Ca_{2+}$ concentration (Fig. 3b). Although calcification is the process most plausibly affected by reduction of SW $Ca_{2+}$ other physiological mechanisms are potentially affected as well and cause downregulation of shell formation. This is included in the revised MS. As an alternative hypothesis, animals attempt to upregulate CS $Ca_{2+}$ concentration to either facilitate calcification or avoid shell dissolution due to calcium carbonate undersaturation in the CS as already discussed in the MS.

Minor comments/corrections: -Line 120-"Finish" should be "Finnish"

Response: Corrected, thanks

-Fig 3 is introduced in the results before Fig 2b,c,d. Further, the paragraph (starting at line 271) regarding results of microelectrode measurements should probably be the last the paragraph of section 3.1. This would move the current, last paragraph of the section (which contains reference to Fig 2bcd) into the appropriate presentation order.

Response: Fig 2a is the first figure of the results line 257-270 and fig 3 is introduced afterwards in the paragraph on the microelectrode measurements. We will move the paragraph on the CS $Ca_{2+}$ measurements as suggested by the reviewer to make the order clearer.

-Line 280-reference to Figure 1, should be Figure 2

Response: Corrected

-It would be helpful to indicate the control treatment levels on Fig 2 since higher and lower [Ca2+] treatments are applied

Response: The measurements were only performed for larvae kept under control and reduced Ca2+ levels. We added this information to the caption.

---

## Editor Comment (EC1) · L.J. de Nooijer (Editor) · 23 Jan 2018

Dear Dr Thomsen and co-authors,

please upload the improved version of your manuscript. I asked the two reviewers to have a look at it together with your replies, which I hope they will do quickly ;).

Sincerely,

Lennart